# Evaluation of Attitude and Knowledge of Endodontic, Pedodontic and SBARD Residents in Saudi Arabia toward Regenerative Endodontics—A National Survey

**DOI:** 10.3390/medicina58040545

**Published:** 2022-04-14

**Authors:** Ali A. Assiry, Mohmed Isaqali Karobari, Niher Tabassum Snigdha, Roshan Noor Mohamed, Syed Nahid Basheer, Mohammed Zameer

**Affiliations:** 1Preventive Dental Science Department, Faculty of Dentistry, Najran University, Najran 55461, Saudi Arabia; 2Conservative Dentistry Unit, School of Dental Sciences, Universiti Sains Malaysia, Health Campus, Kubang Kerian 16150, Kelantan, Malaysia; 3Department of Restorative Dentistry & Endodontics, Faculty of Dentistry, University of Puthisastra, Phnom Penh 12211, Cambodia; 4Center for Transdisciplinary Research (CFTR), Saveetha Dental College & Hospitals, Saveetha Institute of Medical and Technical Sciences University, Chennai 600077, India; 5Paediatric Dentistry Unit, School of Dental Sciences, Universiti Sains Malaysia, Health Campus, Kubang Kerian 16150, Kelantan, Malaysia; dr.nihertabassum@gmail.com; 6Department of Pediatric Dentistry, Faculty of Dentistry, Taif University, Taif 21944, Saudi Arabia; roshan.noor@tudent.edu.sa; 7Department of Restorative Dental Sciences, College of Dentistry, Jazan University, Jazan 45142, Saudi Arabia; snbasheer@jazanu.edu.sa; 8Department of Pediatric Dentistry, Armed Forces Hospital, Jazan 82722, Saudi Arabia; drmohammedzameer@gmail.com

**Keywords:** dentistry, stem cells, awareness, education, postgraduate, regenerative endodontics

## Abstract

*Background and Objectives*: There are no guidelines on the regenerative endodontic procedures that should be followed in dental treatment. Thus, it is essential to learn the knowledge, attitudes, and opinions about regenerative endodontics among dental practitioners, particularly endodontists, as they are the first providers of this type of dental treatment. *Aim*: This study aimed to assess the knowledge and attitudes among residents of the Saudi Board of Advanced Restorative Dentistry (SBARD) in Saudi Arabia. *Methods & Materials*: A questionnaire-based survey was conducted among 128 residents of endodontics, pedodontics, and the SBARD in Saudi Arabia. The questionnaire was distributed to participants using the available electronic measures such as emails and social media. The questionnaire was made by Google Forms, and responses were received directly in the excel sheet in the email. The data were exported as an excel file, coded, and analyzed, and the *p*-value < 0.05 was set as significant. *Results*: More than 50% (74 dentists) of the participants reported receiving continuing education in regenerative treatment, with a significant difference found among specialty (*p* = 0.023). No significant association was found between knowledge, gender, and type of postgraduate study. The vast majority (89.1%) of the participants believed regenerative endodontics should be incorporated into dentistry curricula. About two-thirds (78.2%) of the participants declared their ability to attend future training courses in a regenerative endodontic, whereas 10.9% did not, and 10.9% were unsure. *Conclusions*: The dental practitioners were enthusiastic and interested in regenerative and stem cell treatment. The results also suggested incorporating regenerative treatment into the dental curriculum and signaled the need for more continued training courses.

## 1. Introduction

Regenerative endodontics refers to creating and delivering a newly formed pulp to replace the diseased, missing, or traumatized pulp and surrounding structures with live tissue from the same origin [1]. Different regenerative endodontics procedures have been identified, such as revascularization, apexification, direct pulp capping, partial pulpotomy, apexogenesis, tissue engineering, and stem cell therapy. Several factors can affect successful treatment with regenerative endodontics. Some of these factors are: (i) disinfection of the canal to achieve a clean environment, (ii) apex diameter, and (iii) the patient’s age [2]. Different types of medications and solutions are available as disinfectants. Apex diameter is another factor in the success of regenerative endodontic treatment. An open apex will allow the formation of new tissue by the migration of stem cells into the root canals.

A recent systematic review showed that the influence of the patient’s age on the prognosis of vital pulp therapies (VPT) remains unclear [3]. Several of the reviewed studies included only young patients ranging from 9 to 17 years of age, up to 35 years, reflecting the rationale that the pulp of young patients possesses a higher capacity for recovery. Nevertheless, other studies involving a more comprehensive range of ages from 9 to 69 years did not report an impact of the patient’s age on the outcome of VPT [3]. Revascularization/revitalization techniques aim to increase the length and thickness of the root while maintaining important pulpal characteristics. A tooth that has been successfully revascularized would not require extra treatment [4,5,6,7,8]. Additionally, regenerative endodontic procedures should not be performed on deciduous teeth because of the possibility of impairing the eruption of permanent teeth. Few studies stated some clinical considerations about regenerative endodontics, and no clear guidelines have been developed [9,10]. However, the European Society of Endodontology’s (ESE) Position Statement on Revitalization Procedures and Clinical Considerations for a Regenerative Procedure, both of which were adopted from the American Association of Endodontists (AAE), are available. Such guidelines regarding regenerative endodontic procedures’ ethical use and boundaries are essential for dentists, researchers, and the general population [11]. As a result, it is critical to understand dental practitioners’ knowledge and attitudes about regenerative endodontics, particularly endodontists, who are the first to administer this type of dental treatment [12,13,14,15,16,17].

Endodontics is particularly concerned about advances in endodontic techniques and pedodontics that routinely deal with deciduous teeth where regenerative treatment is contraindicated. The residents of the Saudi board are postgraduate students who should be aware of new technologies developed in dentistry. Few studies have been conducted in this regard, and no study has been conducted in Saudi Arabia. This study aimed to assess the knowledge and attitudes among residents of the Saudi Board of Advanced Restorative Dentistry (SBARD) in Saudi Arabia.

## 2. Materials and Methods

### 2.1. Setting

This study obtained ethical approval from the ethics committee of the Faculty of Dentistry, Najran University, with an ethical approval code (2020/02/0049), 18 May 2020. This questionnaire-based survey was conducted among 128 residents of endodontics, pedodontics, and the SBARD in Saudi Arabia. The data were collected from June to October 2020 with a closed-ended electronic questionnaire to solve the questions of this study. The questionnaire was adopted from previous studies [18,19,20], and no changes have been made to the questions.

### 2.2. Sample Size Determination

The sample size calculation was determined using GPower software (version 3.1) Franz Faul, Universitat Kiel, (Kiel, Germany), using the x^2^ test, to determine that an appropriate sample size for the current study objective would be confirmed with a small to medium effect of size 0.3. Further comparison would be detected reliably, employing the goodness-of-fit test as a statistical test, and an, A priori type of power analysis: compute the required sample size given α, power and effect size with a conventional alpha level of 0.05, and a degree of freedom of 3 with 0.82 power. The obtained sample size was 127 participants.

### 2.3. Questionnaire

The questionnaire consisted of four parts; the first part consisted of questions related to characteristics of the participants, including gender, specialty, type of postgraduate study, level of training, location of postgraduate study, and practice hours per week. The postgraduate studies were of two types, either SBARD residents or master’s students, and the level of training was divided into four levels. Master’s degree students had only level I (R1) and level II (R2) training, whereas SBARD residents had four levels of training, from level I (R1) to level IV (R4) training. The second part of the questionnaire consisted of six questions relating to the participants’ knowledge about regenerative endodontics. The third part consisted of 10 questions relating to the participants’ attitudes toward some issues regarding treatment with regenerative endodontics. The final part consisted of four questions relating to the participants’ agreeability toward practicing and recommending dental treatment with regenerative endodontics. The other three questions were about the possibility of facing cases in clinical practices for which regenerative treatment is indicated.

### 2.4. Administration of the Questionnaire

The electronic questionnaire was distributed to participants using electronic measures such as emails and social media. Responses to questions were received via the website where the questionnaire was created (Google Scholar). The inclusion criteria included Saudi dentists who were SBARD residents or master’s degree holders in endodontics or pedodontics. The summary of the study aims and a protection of privacy was provided at the beginning of the questionnaire, and the respondents were to click a button of acceptance to participate in the study.

### 2.5. Statistical Analysis

The data were exported as an excel file, coded, and analyzed using the SPSS software program version 25 IBM, Stanford University (California, USA). A *p*-value < 0.05 was set as significant. The statistical test used was Pearson’s chi-square used for contingency tables.

## 3. Results

### 3.1. Characteristics of the Sample

There were 128 complete responses received from study participants, and the majority (62.5%) were males. Of the participants, 35.9% were residents of the endodontic department, 35.9% of participants were residents of the pedodontic department, and 28.1% of participants were residents of the SBARD. Most postgraduate students were board graduates, totaling 110 participants (85.9%), and 18 (14.1%) had master’s degrees. About a third of the participants (32%) were in R4 (level 4) of the training, and a slightly lesser number were in R3 (level 3), followed by the number of residents in R2 (level 2) and R1 (level 1). About half of the participants (51.6%) studied in the central region, followed by 32.8% in the western region and only 3.1% in the northern region. Regarding the practiced hours per week, most participants (74.2%) claimed that they practiced more than 20 h per week, whereas 25.8% reported less than 20 h per week. More details about the characteristics of the sample are presented in Table 1.

### 3.2. Knowledge

More than 50% of the participants reported that they had received continuing education in regenerative treatment, with a significant difference by specialty (*p* = 0.023), whereas no significant difference was found according to gender and type of postgraduate study. The majority of participants (86.7%) reported that they had never used umbilical cord or any other type of stem cells, with no significant difference according to the study variables (gender, specialty, type of postgraduate study). About one-third of the participants (38.3%) reported using some regenerative endodontic procedures in their dental practice, and 61.7% reported not using any. There was a significant difference according to specialty (*p* = 0.006), but no significant difference was found according to gender and type of postgraduate study. Half of the participants reported positively to the question related to the healing of non-surgical or root canal treatment, whereas 9.4% had negative answers, and 39.8% did not know. Significant differences in responses were seen according to specialty, with a *p* value < 0.05. Similarly, half of the participants chose “all of the above” for the most valuable regenerative treatment options, with no differences between the study variables. For the optimal treatment of necrotic immature teeth, significant differences were found between the participants according to specialty and type of postgraduate study, but there was no significant difference according to gender (Table 2).

The vast majority (89.1%) of the participants believed that regenerative endodontics should be incorporated into dentistry. No significant differences were found according to the study variables. About 65% of participants thought positively that a stem cell bank would be helpful, with significant differences according to specialty (*p* = 0.007). Nearly half of the participants (51.6%) thought that it will take 0 to 10 years for stem cells to be used in dentistry. A significant difference was also seen according to specialty (*p* = 0.007). One-third of participants thought that dentists need 11 to 20 years to implant new teeth using the regenerative stem cell method. Additionally, a significant difference was found according to specialty (*p* = 0.002). Most residents (61.7%) reported that higher cost was the biggest obstacle to patients accepting stem cell and regenerative treatment. There was a significant difference in gender concerning this question (*p* = 0.020). More than half of the participants (58.6%) believed that regenerative treatment would be better than an implant, 35.9% were unsure, and 5.5% reported that it would not be better. A significant difference was found according to specialty (*p* = 0.042). Most residents (89.1%) reported that regenerative endodontic procedures should be tested on animals before clinical application to patients. There were no significant differences according to the study variables. When the residents were asked about their assessment of regenerative treatment, about half (52.3%) reported that they did not know whether the treatment would be successful, 43% reported that the treatment would be successful, and 4.7% chose that it would be unsuccessful. A significant difference was found according to gender (*p* = 0.001) and specialty (*p* < 0.001). No significant differences were found between participants when asked if professional associations should regulate the use of regenerative endodontic treatment and stem cells. More than 45% of participants believed that the cost of regenerative treatment should be more than the current conventional treatment, 23.4% thought it should be the same, and 7.8% thought that it should be less. A significant difference was found according to specialty (*p* = 0.026) (Table 3).

### 3.3. Attitude

About two thirds (78.2%) of the participants were willing to attend future training courses in regenerative endodontics, 10.9% were not, and 10.9 were unsure. No significant differences were found according to the study variables. Similarly, no significant differences were found between the participants concerning saving teeth and dental tissue for future use in regenerative endodontic treatment, but a significant difference (*p* = 0.005) was found according to gender concerning collecting dental tissues for a stem cell bank. Concerning the reason that would make the residents recommend regenerative treatment for their patients, half of the participants reported that they would recommend regenerative treatment if it was the most effective treatment, one-third (33.6%) of participants would recommend it if it was safe and reliable, 11.7% would recommend it if it was the most cost-effective, and 2.3% reported that they would never recommend regenerative treatment for their patients. No significant differences were found among all the study variables (Table 4).

## 4. Discussion

Regenerative treatment in dentistry has opened the door toward saving vitality and thus extending the lifetime of teeth. Learning the knowledge, opinions, and attitudes of dentists (general practitioners or specialists) is a significant key factor for further guidelines and recommendations regarding regenerative treatment in daily dental practice [11,12]. Endodontists and pedodontists seem to be the first practitioners who would deal with such treatment. This survey aimed to investigate the knowledge and attitudes regarding regenerative endodontic procedures among dental specialists who are most involved in endodontic treatment. This is the first study in Saudi Arabia regarding this issue, to the best of our knowledge.

Most of the participants were on the Board for postgraduate specialty. This may relate to the fact that this type of postgraduate study is preferred in the Saudi dental society, although it is more complicated than a master’s degree or an advanced restorative board. The location of graduation was more frequently in central and western regions, which may correlate to the distribution of graduate educational centers. About two-thirds of participants reported more than 20 h per week of practice. This is consistent with the type of postgraduate study in which practice and rotation shifts are more demanding than in master’s or in board studies. More than half of the participants had taken a continuing education course in regenerative treatment. This result is close to that found by Utneja et al. [20], in which 50.6% of the Indian participants had continuing education. However, this percentage of the current study is higher than that reported by Ez-Abadi et al. [21] and Shah et al. [22], who reported 24.8% and 37% continuing education among Iranian and another group of Indian participants, respectively. These results might be due to the higher availability and accessibility of continuing education in Saudi Arabia than in the other countries. Using the umbilical cord or other types of stem cells was found among 13.3% of participants. This result is similar to that found in a survey conducted in the USA by Epelman et al. [23] in 2009, but it is lower than the results found among the Iranian sample in 2017 [20]. Using some regenerative treatment in dental practice was reported by only 38.3% of participants, with the bulk of these using membranes, scaffolds, or bioactive materials. This result is close to that reported by Ez-Abadi et al. [21] and Deborah et al. [24], but lower than that reported by Epelman et al. [23] and Shah et al. [22]. The majority (65%) of the endodontics and pedodontics SBARD and residents were unaware of prior regenerative endodontic procedures but concerned about their outcomes. According to half of the participants, the regenerative endodontic procedure might be employed in various applications, including peri-radicular bone repair, ongoing root growth in immature teeth, pulp tissue rejuvenation during a root canal, and tooth re-implantation. Similar results were seen in a study conducted among Indian populations [24]. The majority of participants (49%) considered tribiotic paste the optimal treatment for necrotic immature teeth. This is in agreement with a study conducted among south African populations [25]. The administration of calcium hydroxide, followed by MTA apical plugs and backfilling with obturation material, is considered the best treatment for necrotic immature teeth by 17.2% of the participants. This reveals that the residents are untrained in sophisticated regenerative endodontic methods. There is a need for continued training and continuing education programs, from the simplest blood clot revascularization method to the most complex treatment, which involves creating tissue-engineered dental pulp constructs in the laboratory and implanting them into cleaned and shaped root canals related to all treatments that accomplish pulp–dentin regeneration.

Most participants thought regenerative treatment should be incorporated into the undergraduate curriculum and dental education. This is similar to the previous study, suggesting the need to review the current curriculum and include new technologies and methods in learning and practicing [20]. The higher cost of regenerative treatment and stem cells was the biggest obstacle to patients accepting such treatment, as reported by 61.7% of participants, higher than the findings reported by Chitroda et al. [26] and Epelman et al. [23]. The higher cost of such treatment might be due to the more expensive types of equipment and the hard work needed. With the progress of new technology, this cost may decrease significantly. Most of the significant differences between participants’ responses were found concerning their specialty. This may relate to the different quantity of study material in Board and master’s programs, or it might be due to the difference in number from each type. A high percentage of knowledge about regenerative treatment and stem cells may be partially due to the media, hearing about it from friends, or reading about it in academic journals.

On the other hand, this reflects the participants’ positive attitude toward attending continuing courses about regenerative treatment and their willingness to save teeth and dental tissue for stem cell banks. Further research may be conducted to evaluate the dental practitioner’s understanding of the numerous methods, changes, and newer materials available for regenerative endodontic operations. A survey of endodontics and pedodontics residents and instructors in various geographic regions would aid in the development of a better understanding of worldwide knowledge and attitude regarding regenerative endodontics.

### Limitations

Some limitations should be noted in the current study, however. First, electronic-based research relied on readily available electronic measurements, potentially reducing the number of participants. Second, no details about the educational center were taken that would reflect differences in curricula and educational methods. The third limitation is the respondent’s bias. Further studies are recommended among larger samples from different educational centers.

## 5. Conclusions

We conclude that most of the participants had good knowledge and awareness regarding regenerative endodontics, and their attitude was positive toward applying this technique in dentistry. Most of the participants agreed with the importance of including educational material about regenerative endodontics in the curriculum. A significant difference in the level of knowledge was found among different specialties.

## Figures and Tables

**Table 1 medicina-58-00545-t001:** Characteristics of the study participants.

		Frequency	Percent
Gender	Male	80	62.5
	Female	48	37.5
Specialty	Endodontic	46	35.9
	Pedodontic	46	35.9
	SBARD	36	28.1
Postgraduate study	Board	110	85.9
	Master’s degree	18	14.1
Level of training	R1 (level 1)	19	14.8
	R2 (level 2)	30	23.4
	R3 (level 3)	38	29.7
	R4 (level 4)	41	32.0
Location of training	central region	66	51.6
	eastern region	9	7.0
	northern region	4	3.1
	southern region	7	5.5
	western region	42	32.8
Practiced hours	less than 20 h	33	25.8
	more than 20 h	95	74.2

**Table 2 medicina-58-00545-t002:** Responses of the participants to the questions about their knowledge of regenerative treatment.

Question	Items	Response	Gender ** (*p* Values)	Specialty ** (*p* Values)	Type of Postgraduate Study ** (*p* Values)
Received continuing education	no	54 (42.2)	0.079	0.023 *	0.469
	yes	74 (57.8)			
Did you use umbilical cord or any other type of stem cells	no	111 (86.7)	0.403	0.748	0.334
	yes	6 (4.7)			
	unsure	11 (8.6)			
used some types of regenerative endodontic procedures in their dental practice	no	79 (61.7)	0.606	0.006 *	0.562
	yes	49 (38.3)			
After nonsurgical root canal treatment, would the healing of periapical tissues be enhanced by tissue engineering?	no	12 (9.4)	0.053	0.031 *	0.300
	yes	65 (50.8)			
	don’t know	51 (39.8)			
Which of the following regenerative endodontic treatments is the most valuable?	continued root development in immature teeth	21 (16.4)	0.097	0.102	0.864
	healing of periradicular bone	11 (8.6)			
	pulp tissue revitalization within a root canal	30 (23.4)			
	tooth re-implantation	2 (1.6)			
	all of the above	64 (50.0)			
What do you consider to be the optimal treatment for necrotic immature teeth?	calcium hydroxide apexification	12 (9.4)	0.057	0.016 *	0.019 *
	calcium hydroxide application followed by MTA apical plug and backfilling with obturation material	22 (17.2)			
	MTA apical plug and backfill with obturation material	31 (24.2)			
	tribiotic paste and pulpal regeneration	63→49.2			

* Significant difference. ** Two-tailed chi-square test was used to calculate *p* values.

**Table 3 medicina-58-00545-t003:** Responses of the participants to the questions about their beliefs about regenerative treatment.

Question	Items	Response	Gender ** (*p* Values)	Specialty ** (*p* Values)	Type of Postgraduate Study ** (*p* Values)
Do you believe that regenerative endodontics should be incorporated into dentistry?	no	3 (2.3)	0.844	0.504	0.724
	yes	114 (89.1)			
	maybe	11 (8.6)			
Do you think that dental stem cell banking will be useful to be able to regenerate dental tissues?	no	8 (6.3)	0.355	0.007 *	0.990
	yes	84 (65.6)			
	unsure	36 (28.1)			
How many years do you think it will take for some regenerative stem cell therapies to be used in dentistry?	never	6 (4.7)	0.086	0.007 *	0.746
	0–10 years	66 (51.6)			
	11–20 years	45 (35.2)			
	more than 21 years	11 (8.6)			
How many years do you think it will take before dentists are able to implant new teeth grown in a laboratory?	never	11 (8.6)	0.611	0.002 *	0.206
	0–10 years	41 (32.0)			
	11–20 years	44 (34.4)			
	more than 21 years	32 (25.0)			
What do you think would be the biggest obstacle to a patient accepting regenerative dental treatment?	fear of stem cells	19 (14.8)	0.020 *	0.420	0.864
	higher cost	79 (61.7)			
	other reasons	30 (23.4)			
Do you think that regenerative dental treatment will be a better treatment option than tooth implant placement?	no	7 (5.5)	0.865	0.042 *	0.449
	yes	75 (58.6)			
	unsure	46 (35.9)			
Do you think stem cells and regenerative treatments should be tested on animals prior to clinical testing?	no	8 (6.3)	0.741	0.095	0.975
	yes	114 (89.1)			
	unsure	6 (4.7)			
What is your assessment of regenerative dental treatment outcomes?	don’t know	67 (52.3)	0.001 *	0.000 *	0.596
	successful	55 (43.0)			
	unsuccessful	6 (4.7)			
Do you believe that dental professional associations should regulate the use of stem cells and regenerative dentistry?	no	8 (6.3)	0.998	0.488	0.363
	yes	99 (77.3)			
	unsure	21 (16.4)			
What should the cost for regenerative dentistry be?	unsure	28 (21.9)	0.129	0.026 *	0.829
	equal to current treatment	30 (23.4)			
	less than current treatment	10 (7.8)			
	more than current treatment	60 (46.9)			

* Significant difference. ** Two-tailed chi-square test was used to calculate *p* values.

**Table 4 medicina-58-00545-t004:** Responses of the participants to the questions about their attitude towards regenerative treatment.

Question	Items	Response	Gender ** (*p* Values)	Specialty ** (*p* Values)	Type of Postgrad. Study ** (*p* Values)
Would you be willing to attend a training course and/ or continuing education courses to apply regenerative dental treatments?	no	14 (10.9)	0.976	0.546	0.492
	yes	100 (78.1)			
	unsure	14 (10.9)			
Would you be willing to save teeth and dental tissues for future regenerative dental treatment?	no	11 (8.6)	0.258	0.158	0.883
	yes	89 (69.5)			
	unsure	28 (21.9)			
Would you be willing to collect dental tissues for stem cells banks?	no	15 (11.7)	0.005 *	0.666	0.248
	yes	68 (53.1)			
	unsure	45 (35.2)			
What would make you most likely to recommend stem cell and regenerative dental treatments to your patients?	I would never recommend it	3 (2.3)	0.383	0.335	0.813
	if it is the most cost- effective option	15 (11.7)			
	if it is the most effective treatment options	67 (52.3)			
	it is safe and reliable	43 (33.6)			

* Significant difference. ** Two-tailed chi-square test was used to calculate *p* values.

## Data Availability

Any data used in the current article can be provided upon request from the corresponding author.

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
