# Peer review of "Evaluation of Attitude and Knowledge of Endodontic, Pedodontic and SBARD Residents in Saudi Arabia toward Regenerative Endodontics—A National Survey"

_medicina, 2022, doi:10.3390/medicina58040545_

Round 1

Reviewer 1 Report

This article is well structured. The introduction is clear, brief and identifies the main objective of the study. The Materials and methods are well described but don't introduce any kind of novelty. Results are clearly presented, with structured tables that complement the text. The discussion analyses critically the results and compare them to similar studies. Conclusion refers to the identified aim.

However, there are some minor suggestions/corrections to consider: 

  • keyword "dentist" - replace for dentistry
  • line 51 "...are: disinfection of the canal to achieve clean environment..." - refers to more then one factor but only refers one. Please add more factors
  • line 54 "... as they are the first provider of this type dental treatment ..." - review english
  • line 55 - 

    Authors state that “the age of the patient plays an important role in the success of regenerative endodontic treatment”. However, a recent systematic review showed that the influence of the patient’s age on the prognosis of vital pulp therapies (VPT) remains unclear (https://doi.org/10.3390/medicina57060573). Several of the reviewed studies included only young patients with ages ranging from 9 to 17 years, up to 35 years reflecting the rationale that pulps of young patients possess a higher capacity for recovery. Nevertheless, other studies, involving a wider range of ages from 9 to 69 years, did not report an impact of the patient’s age on the outcome of VPT. This ambivalence needs to be presented at introduction.

  • line 60 - 

    As authors state, there are no guidelines for regenerative endodontics, however, they must acknowledge the existence of the European Society of Endodontology (ESE) Position Statement: Revitalization procedures adapted from ESE and Clinical Considerations for a Regenerative Procedure adapted from American Association of Endodontists (AAE).

  • line 69 to 73 "Endodontics are particularly concern about advances in endodontic techniques, in addition to pedodontics who routinely deal with deciduous teeth where regenerative treatment is contraindicated. The residents of Saudi board are postgraduate students who should be
    aware about new technologies develop in dentistry." - Consider moving this sentence to line 65.
  • line 75 - it would be nice to have the number of potential responders (residents of endodontic, pedodontic, and SBARD in Saudi Arabia to whom the questionnaire was sent) 
  • line 84 and 85 - explain R1, R2, R3 and R4
  • lines 87-88; 132 - 133; 183-184; 221-222; 234-235 - review english
  • Table 3, question 6 - "...cells and regenerate treatments..." - regenerative?
  • Line 272-273 and 277-278 - references 6 and 9 are the same 

Author Response

We would like to thank the academic editor and the reviewers for taking their precious time to review this manuscript and give us their comments. We would like to explicitly state that we agree with all the comments as these helped us improve the quality of our paper. We have made a conscious effort to answer all the remarks in the paper as advised by the reviewers and highlighted changes made in yellow for their convenience. Kindly consider these and excuse us for any lapse on our part.

Reviewer 1

This article is well structured. The introduction is clear, brief and identifies the main objective of the study. The Materials and methods are well described but don't introduce any kind of novelty. Results are clearly presented, with structured tables that complement the text. The discussion analyses critically the results and compare them to similar studies. Conclusion refers to the identified aim.

However, there are some minor suggestions/corrections to consider: 

Comments to the Author

  1. keyword "dentist" - replace for dentistry

Author response- Thank you for your insightful comments and suggestions. The keyword “dentist” has be changed to dentistry and added to the revised manuscript.

  1. line 51 "...are: disinfection of the canal to achieve clean environment..." - refers to more then one factor but only refers one. Please add more factors

Author response- Thank you for your insightful comments and suggestions. The other factors has been added to the revised manuscript as per your insightful suggestions.

  1. line 54 "... as they are the first provider of this type dental treatment ..." - review english

Authors response- Thank you for your insightful comments and suggestions. The English has been reviewed as per your insightful suggestion and added to the revised manuscript.

  1. line 55 -  Authors state that “the age of the patient plays an important role in the success of regenerative endodontic treatment”. However, a recent systematic review showed that the influence of the patient’s age on the prognosis of vital pulp therapies (VPT) remains unclear (https://doi.org/10.3390/medicina57060573). Several of the reviewed studies included only young patients with ages ranging from 9 to 17 years, up to 35 years reflecting the rationale that pulps of young patients possess a higher capacity for recovery. Nevertheless, other studies, involving a wider range of ages from 9 to 69 years, did not report an impact of the patient’s age on the outcome of VPT. This ambivalence needs to be presented at introduction.
    Author response-Thank you for your insightful comments and suggestions. The changes has been carried out in the revised manuscript as per your comments.

  1. line 60 -  As authors state, there are no guidelines for regenerative endodontics, however, they must acknowledge the existence of the European Society of Endodontology (ESE) Position Statement: Revitalization procedures adapted from ESE and Clinical Considerations for a Regenerative Procedure adapted from American Association of Endodontists (AAE).

Author response- Thank you for your insightful comments and suggestions. The statement has been added to the revised manuscript as per your insightful suggestions. 

  1. line 69 to 73 "Endodontics are particularly concern about advances in endodontic techniques, in addition to pedodontics who routinely deal with deciduous teeth where regenerative treatment is contraindicated. The residents of Saudi board are postgraduate students who should be aware about new technologies develop in dentistry." - Consider moving this sentence to line 65.

Author response- Thank you for your insightful comments and suggestions. The sentence has been moved as per your insightful suggestion in the revised manuscript.

  1. line 75 - it would be nice to have the number of potential responders (residents of endodontic, pedodontic, and SBARD in Saudi Arabia to whom the questionnaire was sent) line 84 and 85 - explain R1, R2, R3 and R4

Author response- Thank you for your insightful comments and suggestions. The changes has been carried out in the revised manuscript as per your insightful suggestions.

  1. lines 87-88; 132 - 133; 183-184; 221-222; 234-235 - review English

Author response- Thank you for your insightful comments and suggestions. The English revision has been done and added to the revised manuscript as per your insightful suggestions.

  1. Table 3, question 6 - "...cells and regenerate treatments..." - regenerative?

Author response- Thank you for your insightful comments and suggestions. The correction has been carried out in the revised manuscript as per your insightful suggestions.

  1. Line 272-273 and 277-278 - references 6 and 9 are the same 

Author response- Thank you for your insightful comments and suggestions. The citation has been updated as per your insightful suggestion.

Reviewer 2 Report

Abstract:

- Revision Is needed: P-value < 0.05, p= 0.023, please standardised it.

- …...while, 10.9% do not and 10.9 were unsure. (missing %)

Introduction:

- ……these factors are: disinfection of the canal to achieve clean environment. (But the author only listed 1 factor)

- “The aim of this study, therefore, was to assess the knowledge and attitudes regarding regenerative endodontics among endodontic, pedodontic and Saudi Board of Advanced Restorative Dentistry (SBARD) residents in Saudi Arabia” It does not tally with the abstract.

Methodology:

- The study was approved by the ethical committee of College of Dentistry, Najran University (Please list out the ethical approval code for better transparency of data obtained)

- The sample size calculations were conducted, whereas the power calculations were done after data collection of 128 respondents. (It is sceptical to calculate the sample size after the intervention was made, perhaps this could be place at the first part of the methodology)

Results:

- The result part is well-written with detailed information.

Discussion:

Some grammatical errors noted. I do appreciate the authors are trying the explain the reasons behind the results, but it seems that the discussion is superficial at this level. A lot of emphasis has been placed by merely comparing the results with previously reported findings. It will be great if the authors can dive deeper into the context, for instance, “The vast majority of participants thought that regenerative treatment should be incorporated into curriculum and dental education. This is similar to the previous study, suggesting the need for review of the current curriculum and including the new technology of methods in learning and practicing” Do the authors mean undergraduate dental curriculum? As far as I am concerned, most postgraduate curriculum incorporated regenerative endodontics and even in some countries, regenerative endodontics has become part of the undergraduate curriculum. If the authors are referring to the dental curriculum in Saudi, what makes the authors think that it should be incorporated aside from what the respondents answered. Is it the increased in market demand? Stakeholders?

- Aside from this, what are the other recommendations that the authors would like to propose? This will give the readers a better perspective and value towards the study.

- another limitation could be respondent bias and the conclusion need some modification. ‘awareness’ and ‘knowledge’ should be differentiated clearly.

- “they believe in have positive attitude…” please rephrase this.

- The respondents did not agree to incorporated into the curriculum; in fact, they believe that regenerative endodontic should be incorporated into dentistry curriculum. The term should be clearly defined as it brings different meaning.

Overall, I am quite satisfied with the findings, but the discussion part need further exploration and addressing what is happening at the results section to give the readers an in-depth knowledge of what and why the study was conducted and what should be highlighted in the results.

Author Response

We would like to thank the academic editor and the reviewers for taking out their precious time to review this manuscript and give us their comments. We would like to explicitly state that we agree with all the comments as these helped us improve the quality of our paper. We have made a conscious effort to answer all the remarks in the paper as advised by the reviewers and highlighted changes made in yellow for their convenience. Kindly consider these and excuse us for any lapse on our part.

Reviewer 2

  1. Abstract:

 Revision Is needed: P-value < 0.05, p= 0.023, please standardised it. - …...while, 10.9% do not and 10.9 were unsure. (missing %).

Author response- Thank you for your insightful comments and suggestions. The revision has been made in the abstract and added to the revised manuscript as per your insightful suggestions.

  1. Introduction: -

……these factors are: disinfection of the canal to achieve clean environment. (But the author only listed 1 factor).

Author response: Thank you for your insightful comments and suggestions. The other factors has been added to the introduction as per your insightful suggestions.

- “The aim of this study, therefore, was to assess the knowledge and attitudes regarding regenerative endodontics among endodontic, pedodontic and Saudi Board of Advanced Restorative Dentistry (SBARD) residents in Saudi Arabia” It does not tally with the abstract.

Author response- Thank you for your insightful comments and suggestions. The sentence has been updated in the introduction and added to the revised manuscript as per your insightful suggestions.

  1. Methodology:

- The study was approved by the ethical committee of College of Dentistry, Najran University (Please list out the ethical approval code for better transparency of data obtained)

Authors response: Thank you for your insightful comments and suggestions. The ethical code has been mentioned in the methodology in the revised manuscript as per your insightful suggestions.

- The sample size calculations were conducted, whereas the power calculations were done after data collection of 128 respondents. (It is sceptical to calculate the sample size after the intervention was made, perhaps this could be place at the first part of the methodology).

Author response- Thank you for your insightful comments and suggestions. Apologies, the sample size calculations is placed at the first part of the methodology as per your insightful suggestions in the revised manuscript.

  1. Results:

- The result part is well-written with detailed information.

Author response- Thank you for your insightful opinion.

  1. Discussion:

Some grammatical errors noted. I do appreciate the authors are trying the explain the reasons behind the results, but it seems that the discussion is superficial at this level. A lot of emphasis has been placed by merely comparing the results with previously reported findings. It will be great if the authors can dive deeper into the context, for instance, “The vast majority of participants thought that regenerative treatment should be incorporated into curriculum and dental education. This is similar to the previous study, suggesting the need for review of the current curriculum and including the new technology of methods in learning and practicing” Do the authors mean undergraduate dental curriculum? As far as I am concerned, most postgraduate curriculum incorporated regenerative endodontics and even in some countries, regenerative endodontics has become part of the undergraduate curriculum. If the authors are referring to the dental curriculum in Saudi, what makes the authors think that it should be incorporated aside from what the respondents answered. Is it the increased in market demand? Stakeholders?

Authors response: Thank you for your insightful comments and suggestions. The authors meant for undergraduate dental curriculum; the sentence has been updated in the manuscript as per your insightful suggestions.

- Aside from this, what are the other recommendations that the authors would like to propose? This will give the readers a better perspective and value towards the study.

Authors response: Thank you for your insightful comments and suggestions. The recommendations are added to the revised manuscripts as per your insightful suggestions.

- another limitation could be respondent bias and the conclusion need some modification. ‘awareness’ and ‘knowledge’ should be differentiated clearly. - “they believe in have positive attitude…” please rephrase this.

Authors response: Thank you for your insightful comments and suggestions. The changes has been carried out in the revised manuscript as per your insightful suggestions.

- The respondents did not agree to incorporated into the curriculum; in fact, they believe that regenerative endodontic should be incorporated into dentistry curriculum. The term should be clearly defined as it brings different meaning.

Authors response: Thank you for your insightful comments and suggestions. The changes has been carried out in the revised manuscript as per your insightful suggestions.

Overall, I am quite satisfied with the findings, but the discussion part need further exploration and addressing what is happening at the results section to give the readers an in-depth knowledge of what and why the study was conducted and what should be highlighted in the results.

Authors response: Thank you for your insightful opinion. The correction has been carried out in the revised manuscript as per your insightful suggestions.

Reviewer 3 Report

Dear Authors,

Please find below some recommendations I believe would improve the quality of the manuscript.

  1. In lines 49 and 50, you mentioned “several factors that affect successful treatment”, however you only mentioned one. Please add more factors.

  1. In lines 130 and 131, please specify if there were any differences between endodontics and pedodontics specialties regarding the use of these treatments.

  1. The discussion is short and only 4 or 5 questions are compared to other studies. Please, add more comparisons of additional question results with other studies available if possible.

Author Response

We would like to thank the academic editor and the reviewers for taking out their precious time to review this manuscript and give us their comments. We would like to explicitly state that we agree with all the comments as these helped us improve the quality of our paper. We have made a conscious effort to answer all the remarks in the paper as advised by the reviewers and highlighted changes made in yellow for their convenience. Kindly consider these and excuse us for any lapse on our part.

Reviewer 3

Dear Authors,

Please find below some recommendations I believe would improve the quality of the manuscript.

  1. In lines 49 and 50, you mentioned “several factors that affect successful treatment”, however you only mentioned one. Please add more factors.

Author response- Thank you for your insightful comments and suggestions. The other factors has been added to the revised manuscript as per your insightful suggestions.

  1. In lines 130 and 131, please specify if there were any differences between endodontics and pedodontics specialties regarding the use of these treatments.

Author response- Thank you for your insightful comments and suggestions. There was no difference

  1. The discussion is short and only 4 or 5 questions are compared to other studies. Please, add more comparisons of additional question results with other studies available if possible.

Author response- Thank you for your insightful comments and suggestions. The correction has been carried out in the revised manuscript as per your insightful suggestions.

Round 2

Reviewer 2 Report

The authors have addressed each concern raised by the reviewer appropriately. One minor suggestion is to add relevant references to the point below (Introduction):

"Several factors can affect the successful treatment with regenerative endodontics; some of these factors are ...... Apex diameter iii) Patient's age."

Author Response

We would like to thank the academic editor and the reviewers for taking out their precious time to review this manuscript and give us their comments. We would like to explicitly state that we agree with all the comments as these helped us improve the quality of our paper. We have made a conscious effort to answer all the remarks in the paper as advised by the reviewers and highlighted changes made in yellow for their convenience. Kindly consider these and excuse us for any lapse on our part.
